# Residential Demand Response Strategy Based on Deep Deterministic Policy Gradient

**Chunyu Deng** [1,2,*] **and Kehe Wu** [1]

1   School of Control and Computer Engineering, North China Electric Power University, No. 2 Beinong Road, Changping District, Beijing 102206, China; wkh@ncepu.edu.cn
2   China Electric Power Research Institute, No. 15, Qinghe Xiaoying Road, Beijing 100192, China
*   Correspondence: dengchunyu@epri.sgcc.com.cn

**Abstract:** With the continuous improvement of the power system and the deepening of electricity market reform, the trend of users' active participation in power distribution is more and more significant. Demand response has become the promising focus of smart grid research. Providing reasonable incentive strategies for power grid companies and demand response strategies for customers plays a crucial role in maximizing the benefits of different participants. To meet different expectations of multiple agents in the same environment, deep reinforcement learning was adopted. The generative model of residential demand response strategy under different incentive policies can be trained iteratively through real-time interactions with the environmental conditions. In this paper, a novel optimization model of residential demand response strategy, based on a deep deterministic policy gradient (DDPG) algorithm, was proposed. The proposed work was validated with the actual electricity consumption data of a certain area in China. The results showed that the DDPG model could optimize residential demand response strategy under certain incentive policies. In addition, the overall goal of peak load-cutting and valley filling can be achieved, which reflects promising prospects of the electricity market.

**Keywords:** demand response; deep reinforcement learning; deep deterministic policy gradient; power consumption strategy optimization





## 1. Introduction

Opening the electricity market is the main intent of the new electric power system reform in China. According to the statistics, there are more than 3000 electricity retailers in China. As the number of power retailers increases, depending on flexible price/incentive mechanism users can stimulate the demand response potential of electricity sales companies and the wholesale smart grid to improve the efficiency. Moreover, the proportion of residential electricity consumption in China is constantly growing in recent years. For example, the peak load of residential electricity consumption in Jiangsu province was greater than 1/3 span in the summer of 2020. Besides, the rapid popularization of smart home products has greatly enhanced the potential of residents' demand response. Therefore, the optimization of large-scale smart residential users' electricity usage strategy in a complex market environment has important theoretical and practical values.

In order to effectively promote clean energy consumption, the Electric Internet of Things (EIOT) technology needs to be introduced. However, the demand for adjustable capacity has gradually expanded, due to the unbalanced outputs of clean energies connected to the power grid. One of the most effective strategies to solve the uncertain integration of large-scale renewable energy in the grid is aggregating controllable loads on the user side in real-time [1]. With the introduction of the market concept into the power system, attention paid to the demand side and the interactive demand of the power system have been raised to a new level, and participants in the market showed a significant trend of diversification [2]. At the same time, more and more emerging market players and

load aggregators appear in front of market participants [3]. The residential load has the characteristics of small units, large amounts, and scattered locations. The main difficulties for electricity sales to optimize power consumption operations are how to aggregate and manage more residents to participate in the grid interactions in a more comfortable way. In general, this includes the optimal decision of electricity purchase and sale and feasible load scheduling plans for users.

To optimize the operation of electricity sales companies to maximize profits, a well-designed interactive model is indispensably considering the load profiles of residents and the decision-making of electricity trading based on the demand response in the electricity market. At present, some exploratory studies have been carried out. Reference [4] applied the utility function to build the residents' demand response model, and electricity sales companies motivate users to adjust their electricity consumption patterns by implementing load-cutting compensation. Reference [5] integrated residential air conditioning load into the direct control load for dispatching, and an optimized operating model based on the combination of the distributed power supply and the distribution network constraints is established. Reference [6] designed a response framework for air conditioning load, which allocated incentive costs based on user engagement to reduce the impact on user comfort. Reference [7] provided a novel strategy that users could adjust the load operations to realize the demand response on multi-smart homes with regional real-time electricity price predicting. In Reference [8], load aggregators divided residential loads into three categories according to their operation characteristics. Each category of the load is scheduled by grouping aggregations to maximize its benefits in the day-before market. In reference [9], constraints of resident load were considered. The authors established a multi-user intelligent electricity scheduling model and presented a particle swarm optimization algorithm. Reference [10] adopted quadratic utility function modeling for residential users and built a multi-user demand response model for single-period multi-seller electricity companies by applying a two-layer master-slave game. Reference [11] proposed a distributed control model based on the game theory framework to solve the problem of demand response under many flexible loads in the market environment. Reference [12] proposed a distributed optimization framework for intelligent electricity consumption in residential areas. In the previous references, researchers basically tried to obtain more advanced models to describe the real competitive environment of the instant electricity trading market, which equates with a multi-objective optimization task. However, because of the great difficulties in centralized optimization, the aforementioned studies that depended on grouping aggregation and utility function for describing the residential characteristics, cannot be applied in appliance-level objectives. Some other studies focused on residential cases with lightweight models, but it could not be fully expanded to proper scale communities. References [10–12] solved the contradiction between the accuracy and scale of the residential demand response model based on the game theory using the distributed optimization characteristics. Yet, it needs to prove the existence of a pure strategic Nash equilibrium solution, which limits the possible applications of game theory in residential demand response.

In summary, giving back to customers involved in grid interactions by reducing their power consumption costs might have great significance in attracting large-scale customers to participate in demand response, allowing power companies to obtain large-scale adjustable loads. Therefore, making a reasonable incentive electricity price and achieving a balance between expanding the load scale and reducing the incentive cost are the key issues to be solved. A strategy optimized model based on deep deterministic policy gradient (DDPG) algorithm is proposed to solve the retail-price making problem for the service provider, and finally verified the effectiveness of the model by using the real power consumption data. This paper is organized as follows. Section 2 gives a brief review of residential demand response and deep reinforcement learning. Section 3 gives the residential demand response model and Section 4 shows the strategy optimized by DDPG and simulations are conducted. Section 5 analyzes the results, and conclusions are given in Section 6.

## 2. Related Work

### 2.1. Residential Demand Response

With the continuous increase in new forms of energy consumption, such as electric vehicles and lithium battery storage, it is possible to use incentive measures to guide users to actively participate in load regulation, such as peak shaving and valley filling. The research aim of residential demand response strategy optimization is to reasonably determine the power consumption involved in demand response based on the market demand and the operator's incentive policy in order to achieve the maximum benefit.

Reference [13] analyzed the variations of load behaviors on the residential side in detail under the incentives of different electricity prices. The authors discussed how the injected power of the distribution network is regulated by demand response resources after adding highly volatile distributed generation into the distribution network and studied the contribution of demand response in smoothing the fluctuation of distributed generation and reducing the load peak. Reference [14] studied the scheme of reducing household electricity consumption and promoting the two-way interactions between the user side and the grid side. Reference [15] paid more attention to automatic demand response, analyzed the obstacles existing in the implementation of automatic demand response in China, and figured out the potential and development direction of China's automatic demand response. Reference [16] proposed the design principles of the demand response market and the constituent elements of the demand response project. In addition, it classified the types of demand response markets and demand response projects. References [17–19] presented a demand response strategy based on the dynamic price in the smart grid environment. Reference [20] proposed the identification model of user interaction degree and solved the problem of processing the inflection point of a piecewise linear response curve through repeated correction of historical data. Tan, et al. [21] gave an optimization model for the design of time-of-use (TOU) price on the generation side considering the real-time grid electricity in different peak valley peacetime periods. Reference [22] designed an optimal decision-making model of TOU price considering the satisfaction of consumers' electricity consumption behavior. Reference [23] proposed a bilateral price linkage electricity price model based on the field data of municipal power companies and obtained the optimal TOU price for peak load shifting and valley filling through simulation analysis. Wei, et al. [24] studied the peak-to-valley electricity price of high load energy users, took the wind power consumption capacity into account, established a corresponding peak-valley electricity consumption model with continuous division of peak-to-valley periods, and finally established the peak-to-valley electricity price decision-making model with the goal of increasing the wind power consumption capacity of the system during the valley period. Through decoupling time division and electricity price optimization, a two-step optimization method was designed to build the comprehensive optimization model of peak-to-valley period and electricity price. Reference [25] presented the optimal operation scheme for household user side micro-grid considering the TOU price and demand response. Reference [26] devised the optimization model for large consumers' direct wind power consumption under TOU price. Reference [27] proposed the multi-objective optimal scheduling of electric vehicle charging and discharging based on TOU price. Fan et al. [28] provided a price optimization strategy for the access time of photovoltaic power generation in the smart grid. Overall, the studies of demand response strategy usually rely on the mathematical models which need more work on the models of participants with this type of formula. It is necessary to look at data-driven methods by using artificial intelligence technology.

### 2.2. Deep Reinforcement Learning

Reinforcement learning, supervised learning and unsupervised learning constitute the core machine learning section of artificial intelligence technology. Supervised learning is learning from a marked training set. The characteristics of each sample in the training set can be regarded as a description of the scene, and its label can be regarded as the correct

action that should be performed. The purpose of unsupervised learning can be defined as finding the hidden rules from an amount of unlabeled samples.

Reinforcement learning runs in a specific situation with the process of trying and choosing which kind of action can get the maximum rewards. The difference between reinforcement learning and other machine learning algorithms is that there is no supervisor in the learning process, where the interactions between actions and related feedbacks occur in this process. Therefore, decision-making behavior will affect a series of states and rewards significantly.

Deep reinforcement learning combines deep learning skills with reinforcement learning. Deep learning is a neural network with hidden multi-layer. Its basic theory is to enhance low-level features through multi-layer network structure and nonlinear transformation and to discover the distinguished representation of the target by acquiring abstract and more discriminative high-level representation. In recent years, deep learning skills, such as the most popular supervised learning, have been successfully applied in smart grids through load forecasting [29], price peak prediction [30], optimization of battery storage [31], etc. Reinforcement learning comes from the behaviorism theory in psychology which produces the habitual behavior that can obtain the maximum benefit under the guide of the reward or punishment scheme given by the environment. Deep reinforcement learning transforms the traditional value iteration into the training of the neural network, breaks the limitation of state dimension and discrete action in the classical reinforcement learning, and uses a deep neural network with a strong fitting ability to establish strategy function [32,33]. Deep reinforcement learning technology has been widely used and plays an important role in games, robot control, dialogue system, autonomous driving, and other fields [34–40].

Deep reinforcement learning algorithm is described and classified in reference [41], which can be divided into two basic methods based on value function and policy gradient. Typical classifications of deep reinforcement learning algorithms are shown in Table 1, among which deep Q-learning (DQN), advantage actor-critic (A3C), DDPG, and Distributed proximal policy optimization (DPPO) are the most representative deep reinforcement learning models at present.

**Table 1.** The classification for typical reinforcement learning (RL) and deep reinforcement learning (DRL) models.

| Classification | Reinforcement Learning | Deep Reinforcement Learning |
|---|---|---|
| Based on value function | Q-learning Sarsa | Deep Q Network, DQN |
| Based on policy gradient | Reinforce algorithm | Deterministic policy gradient, DPG Deep deterministic policy gradient, DDPG |
| | actor–critic, AC | Trust region policy optimization, TRPO Asynchronous advantage actor-critic, A3C Distributed proximal policy optimization, DPPO |

The DDPG algorithm is a combination of the actor–critic framework and the DQN algorithm. The experience playback mechanism and a separate target network are used to reduce the correlation between the data set and increase the stability and robustness of the algorithm. In addition, considering the problem that the continuous action space is easy to fall into the local optimum, the design of the "noise" item is added, and the exploration efficiency is improved by strengthening the random behaviors in the action strategy. Both the actor and critic dual networks in the DDPG algorithm adopt the target network and evaluate the network design. During the training process, in order to increase the stability of the algorithm learning process, only the parameters of the estimated network need to be updated, and the training steps are separated and copied to the target network.

A systematic review of artificial intelligence and machine learning methods used in DR is reported in [42]. Reinforcement learning [43] especially deep reinforcement learning [44], such as advantage actor-critic (A3C) and deep Q-learning (DQN) [45] are

promising approaches for strategy optimization of residential demand response. The DDPG model had not been used to solve the problems of residential demand response, especially not to be used in generating retail prices for the service provider.

## 3. Research on Demand Response Model

When there is a power shortage in the real-time balance market, the power generator increases the output, and the system operator bears the cost of system adjustment. Therefore, by introducing demand response into real-time market balance, system operators subjectively design reasonable demand response strategies for various demand responses, guide demand-side responses, and reduce the system adjustment costs. To make reasonable demand response strategies, it is necessary to consider the response characteristics of different market participants and establish the demand response model which is showed in Figure 1. It consists of a power grid company, service provider, customers, and other market participants.

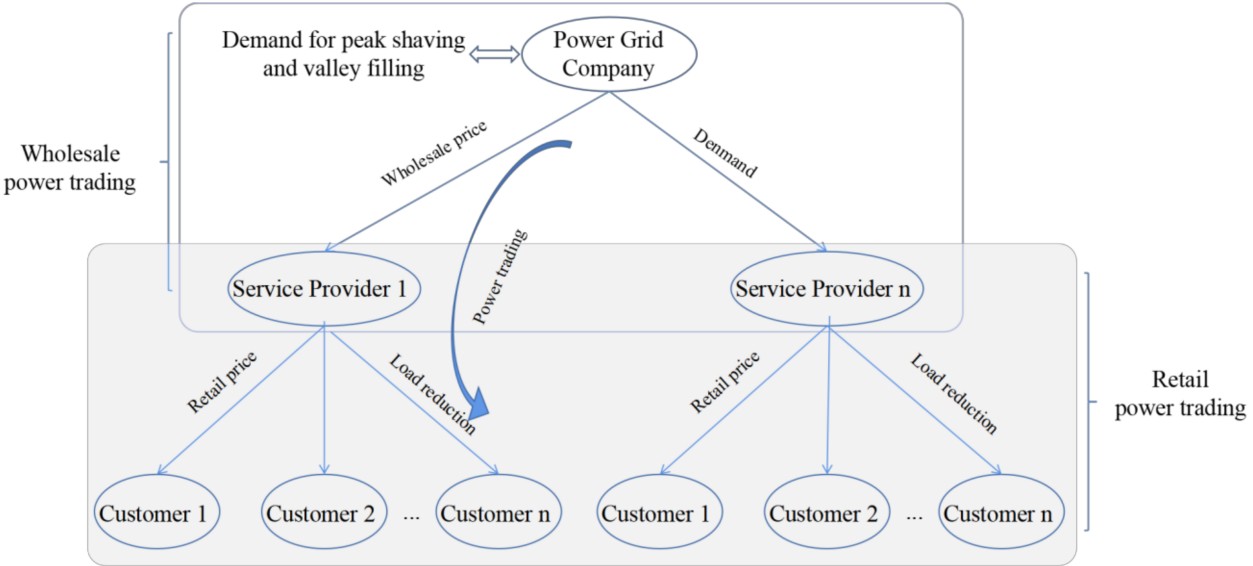

**Figure 1.** Power trading model considering the service provider.

The power grid company usually formulates the wholesale price and supplies the power to the service provider. Power grid companies would evaluate the demand from a large amount of metering and load forecasting data. Because of the electric reformation, some authorities should be delivered to the service provider such as the price ensure of electricity.

The service provider is between the power grid company and customers, operating an incentive-based demand response program with its customers that encourages the customers to sell demand reduction in exchange for a lower retail price. Here we give the contact between retail price and customers' power reducing $\Delta E_{n,t}$ which is described as follows:

$$\Delta E_{n,t} = E_{n,t} \cdot \xi_t \cdot \frac{P_{n,t}^{retail} - P_{min}}{P_{min}} \tag{1}$$

In Equation (1), $E_{n,t}$ indicates the energy demand of customer $n$ at time $t$, and $\xi_t$ is the elasticity coefficient at time $t$ that denotes the ratio of energy demand change to the retail price. In the paper, $P_{n,t}^{retail}$ is the retail price at time $t$ for customer $n$, $P_{min}$ is set as the wholesale price which usually is fixed over a period and should be announced by the power grid company.

Customers are enrolled in the demand response program, including large industrial users, load aggregators, end-users, and other market participants. When they are informed

of the retail price by the service provider, customers try to maximize their benefits by decreasing their power consumption to get a lower retail price. the electricity benefit is the sum of the electricity costs for actual power consumption which is described as follows:

$$B_{benefit} = \sum_{t=1}^{T} \left( P_{n,t}^{retail} - P_t^{wholesale} \right) \cdot \Delta E_{n,t} \tag{2}$$

In Equation (2), $P_t^{wholesale}$ is the wholesale price, $P_{n,t}^{retail}$ is the retail price.

Otherwise, a dissatisfaction cost function is used in this paper which is described as follows:

$$\varphi_{n,t}(\Delta E_{n,t}) = \frac{\mu_n}{2}(\Delta E_{n,t})^2 + \omega_n(\Delta E_{n,t}) \tag{3}$$

In Equation (3), $\mu_n$ and $\omega_n$ are customer-dependent parameters, where $\mu_n$ is a customer label that varies among different power usage. $\omega_n$ is an auxiliary coefficient of the dissatisfaction cost function, and the details are described in [46].

## 4. Residual Demand Response Strategy Based on Deep Deterministic Policy Gradient

### 4.1. The Optimal Model of Power Consumption Strategy

An actual situation that the electric operators often confront is the task of regulating the load. If the power suppliers cannot satisfy the power demand, it will cause economic losses and affect economic developments. In the past, regulating the output of generating units is the only effective approach to address this issue. Once the output limit is exceeded, the normal power loads of customers in the country and other unimportant loads would be forced to cut off. This simple strategy seriously affects the power quality and satisfaction of customers. With the development of China's economy and the continuous improvement of people's living standards, the frequency of power blackout has become an important quality index to measure the level of power consumption. The simple load shedding strategy in the past can no longer meet the current demand. Therefore, it is an ideal choice to use incentives to guide customers to actively participate in the load regulation with the participation of the service provider.

As shown in Figure 2, the power grid company is making the wholesale price based on a load forecasting curve and related policy. Instead, the peak shaving and valley filling demand should relay to service providers. Then, the service provider would make the retail price for customers. It is worth noting that both the customer and the service provider need to survive by maximizing their benefits, so retail price making is an optimization question. In other words, the service provider should economically formulate a reasonable retail price to realize the load regulation and maximize the benefits. Meanwhile, customers hope that costs could be minimized according to the existing retail price and the adjustable load.

Combined with the descriptions in Section 3, several settings about this model can be obtained:

(1) At any time, the total amount to be regulated in load regulation should be less than the adjustable load of customers and the service provider hope the rate of actual power consumption and adjustable load will be greater.

(2) We set the retail price in a range that is greater than or equal to the wholesale price, and less than or equal to the twice wholesale price. If the retail price is lower than the wholesale price, or the retail price is too higher that customers will no longer wish to buy the electricity, and the survival of service providers will be the biggest problem.

(3) From the perspective of residential electricity demand, it can be assumed that the total amount of consumers who wish to actively participate in load regulation is no more than 50% of their normal load. The number of consumers who actively participate in demand response would be influenced by the retail price.

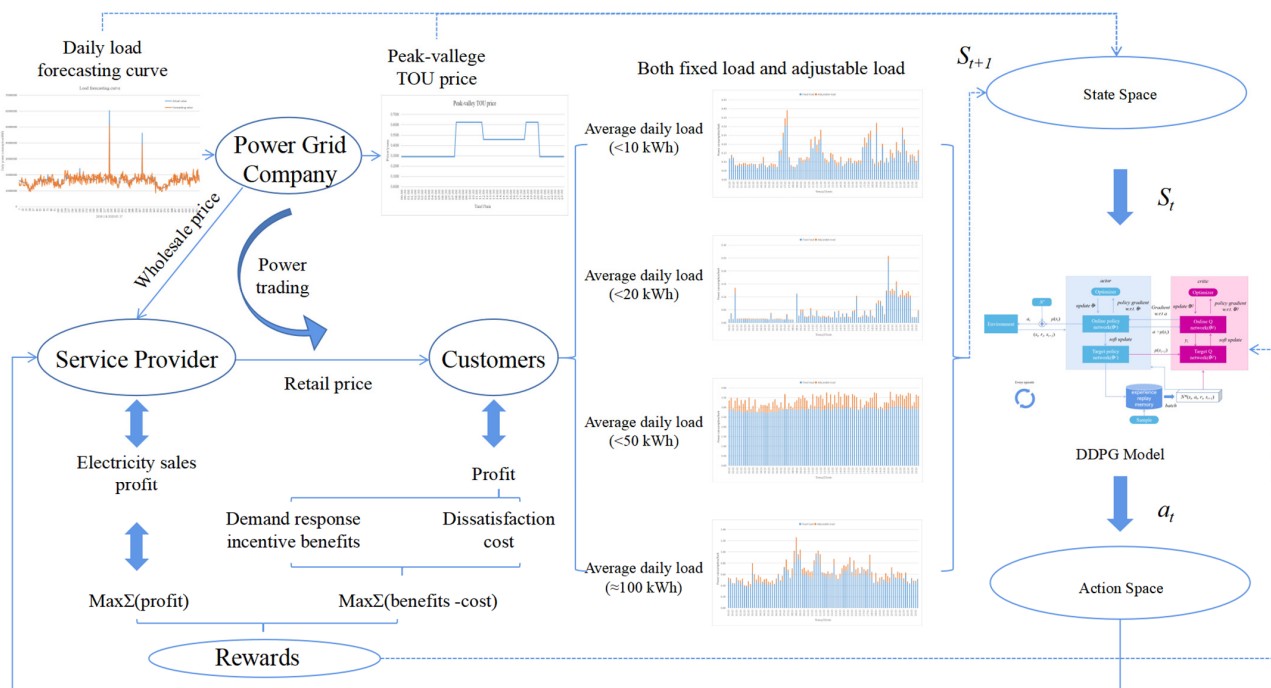

**Figure 2.** The optimal model of power consumption strategy based on DDPG.

### 4.2. Deep Deterministic Policy Gradient Algorithm

Our proposed DDPG is categorized in the deep reinforcement learning area, depending on actor-critical framework and DQN algorithm. In both actor and critical, there are target net and eval net as the baseline for respective sections. In the training process, only the parameters of the estimation network need to be updated, and the parameters of the target network are directly copied by the estimation network in each loop, as shown in Figure 3. The entire process of DDPG is illustrated in Table 2. At first, network parameters are required to be initialized. Then the related actions are activated, agents will be responded to in the corresponding states, and the instant rewards could be acquired.

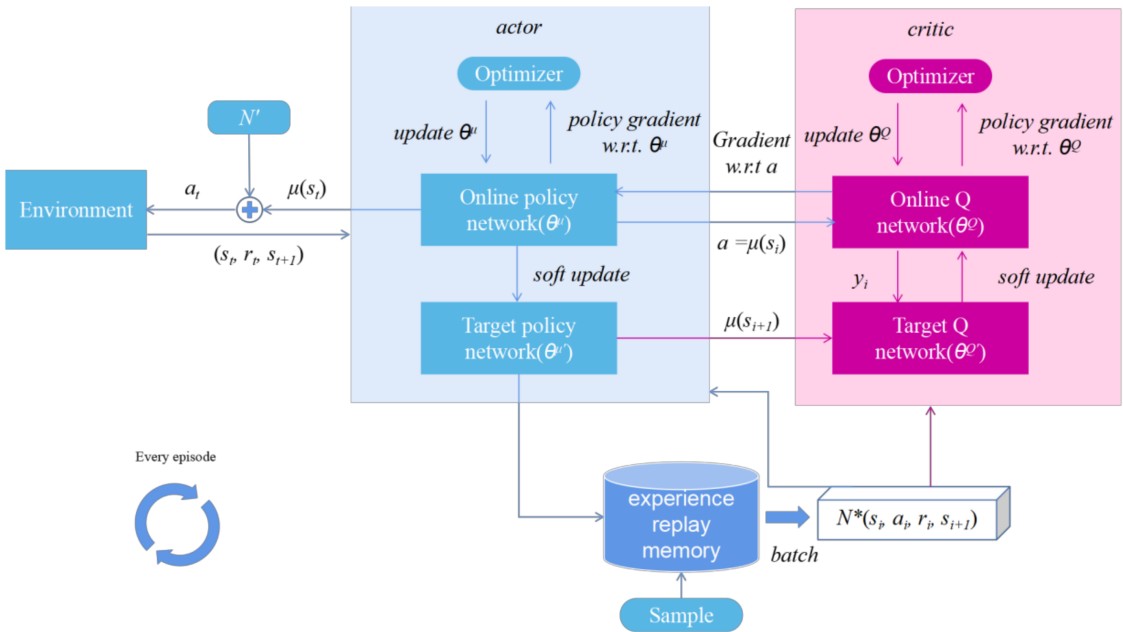

**Figure 3.** The principle of network parameters updating for DDPG algorithm.

**Table 2.** The DDPG Algorithm.

| DDPG Algorithm |
| --- |
| 1         Random initialize network parameters of neural network $\theta^Q$ and $\theta^C$ |
| 2          Initialize target network parameters $\theta^{Q'} \leftarrow \theta^Q$ and $\theta^{\mu'} \leftarrow \theta^\mu$ |
| 3         Initialize playback unit $R$ |
| 4        For episode = 1, max_episode do: |
| 5         Initialize the random noise $N$ for the action to improve the exploration rate |
| 6         Read initialization status $S_1$ |
| 7        For t = 1, T do: |
| 8          Select action according to the current strategy and random noise $a_t = \mu(s_t|\theta^\mu) + N_t$ |
| 9          Executive action $a_t$, get reward $r_t$ and new state $s_{t+1}$ |
| 10         Save the action state sequence $(s_t, a_t, r_t, s_{t+1})$ to the playback unit $R$ |
| 11         Randomly sampled the number of action state sequences with the same batch size from the revisit unit $(s_t, a_t, r_t, s_{t+1})$ |
| 12         Set $y_i = r_i + \gamma Q'\left(s_{i+1}, \mu'\left(s_{i+1}\middle|\theta^{\mu'}\right)\middle|\theta^{Q'}\right)$ |
| 13         Update critical network parameters according to LOSS : $Loss = \frac{1}{N}\sum_i (y_i - Q(s_i, a_i|\theta^Q))^2$ |
| 14         Update actor-network parameters according to the policy gradient: <br> $\nabla_{\theta^\mu} J \approx \frac{1}{N}\sum_i \nabla_a Q(s, a|\theta^Q)\big|_{s=s_i, a=\mu(s_i)} \nabla_{\theta^\mu} \mu(s|\theta^\mu)\big|_{s_i}$ |
| 15         Update target network parameters: <br> $\begin{cases} \theta^{Q'} \leftarrow \tau\theta^Q + (1-\tau)\theta^{Q'} \\ \theta^{\mu'} \leftarrow \tau\theta^\mu + (1-\tau)\theta^{\mu'} \end{cases}$ |
| 16       End |
| 17    End |

Loss function with critical network learning as follows:

$$\begin{cases} y = r + \gamma \max_{a'} \overline{Q}^*(s', a') \\ L(\theta) = E_{s,a,r,s'}\left[(Q^*(s, a|\theta) - y)^2\right] \end{cases} \tag{4}$$

In Equation(4), where network hyper-parameters are estimated according to the critical value. *a* represents the action from the network estimated by the actor, and y is the Q value of the target network. Because the deterministic strategy is adopted, the greedy algorithm is no longer used to select the action when calculating the target Q value. In general, the training of critical estimation network is still derived from the target Q value and the square loss of the estimated Q value. The estimated Q value is obtained by inputting the current state s and the action *a* of the actor estimation network into the critical estimation network, while the target Q value is acquired according to the reward R. The Q value that comes from the next state and the action from the actor target network with the discount factor, then the sum of sub-addition is obtained.

The actor network updates the parameters based on the deterministic strategy according to Equation (5).

$$\nabla J(\theta) = E_{sD}[\nabla_\theta \mu_\theta(a|s)\nabla_a Q^\mu(s, a)|a = \mu_\theta(s)] \tag{5}$$

Assuming that the action space is continuous, two different actions are output for the same state. Two feedback values are obtained, based on the state estimation network. More rewards can be acquired by executing actions. The idea of a strategy gradient is to ensure an appropriate probability for obtaining a larger Q value.

### 4.3. Key Parts of the Model

State space, action space, and reward function are three key parts of the model which directly affect performances of optimal strategy based on DDPG.

### 4.3.1. State Space

State space *S* is the observation of the environment which could be changed by the action. In the optimal model, the parameters $\Delta E_{n,t}$ are illustrated as Equation (6) which is the demand reduction offered by.

$$s \in S(F_t(load), P_t^{wholesale}, \Delta E_{n,t}) \tag{6}$$

In Equation(6), $F_t(load)$ is the forecasting data for total load, $P_t^{wholesale}$ is the wholesale price formulated by the power grid, $\Delta E_{n,t}$ is the actual power consumption of customer n at time *t*.

### 4.3.2. Action Space

Action space describes the collection of behaviors and strategies which would change the status of the environment. The target of the optimal model is to improve the devotion of customers to increase the amount of active participation for load regulation. In this paper, the retail price $P_{n,t}^{retail}$ is chosen as the action behavior which can be described in Equation (7).

$$a \in A(P_{n,t}^{retail}) \tag{7}$$

### 4.3.3. Design of Reward Function

Design of the reward function is an important task for deep reinforcement learning. In this model, the reward system is described in Equation (8) and divided into three parts.

$$r(s_{n,t}, a_{n,t}) \in R(\Delta E_{n,t}, P_{n,t}^{retail}) \tag{8}$$

Part one:

$$R_1 = \begin{cases} +100, success \\ -1000, fail \end{cases} \tag{9}$$

In Equation (9), if the retail price for the service provider is during the $[P_{min}, P_{max}]$ and the benefits of the demand response outweigh the costs, the strategy could be regarded as successful. The reward is increased by 100; otherwise, it should be regarded as failure and the reward is reduced by 1000. In the paper, we set $P_{min}$ to be the wholesale price and the $P_{max}$ to be twice the wholesale price.

Part two:

$$R_2 = \sum_{n=1}^{N} \sum_{t=1}^{T} (P_{n,t}^{retail} - P_t^{wholesale}) \cdot \Delta E_{n,h} \tag{10}$$

In Equation (10), the reward function is the sum of economic benefits for the service provider. It includes the electricity sales and benefits for completing the demand response task, which usually is provided by the power grid company or government.

Part three:

$$R_3 = \sum_{t=1}^{T} [\rho \left( P_{n,t}^{retail} - P_t^{wholesale} \right) \cdot \Delta E_{n,t} - (1-\rho) \cdot \varphi_{n,t}(\Delta E_{n,t})] \tag{11}$$

In Equation (11), the reward function is the sum of the cost for customers. It includes the electricity cost and the dissatisfaction cost as in Equation (3). The parameter $\rho$ is the weight that describes the classification for customers. The value of $\rho$ is greater, and the customer should pay more attention to the benefit for demand response. Otherwise, the customers should pay more attention to their discomfort when their electricity consumption is reduced.

## 5. Result Analysis

### 5.1. Peak-Valley Time of Use (TOU) Price

In this paper, a peak-valley TOU price is used for industrial and commercial electricity consumption as shown in Figure 4. The valley period is from 22:00 p.m. to 8:00 a.m. of the next day, the normal period is from 12:00 a.m. to 18:00 p.m., and the peak period is from 8:00 a.m. to 12:00 a.m. and from 18:00 p.m. to 22:00 p.m.

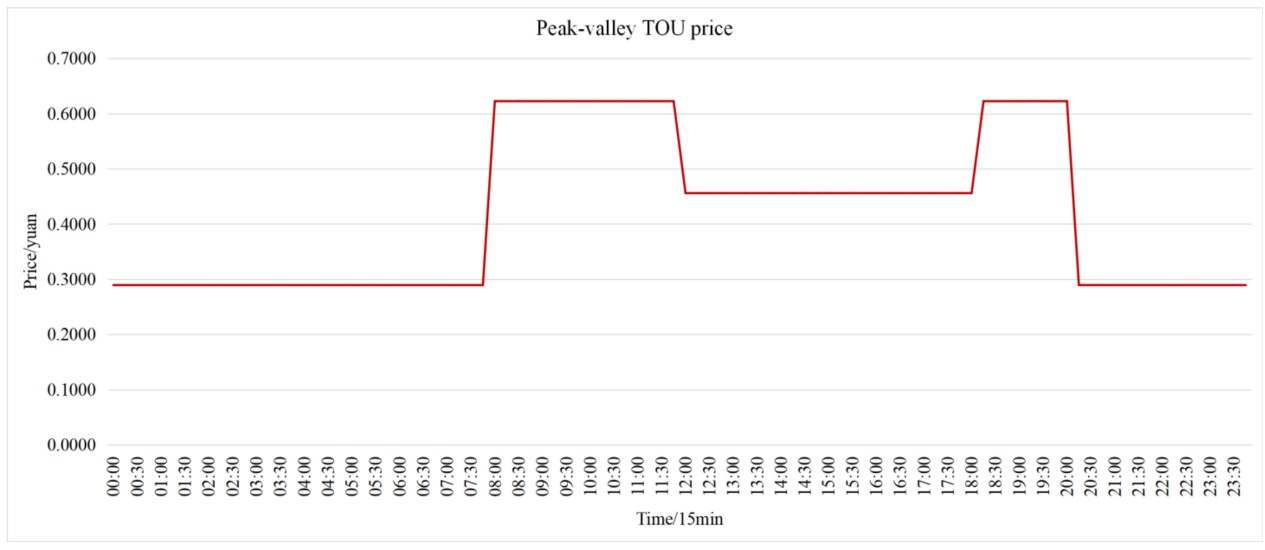

**Figure 4.** Peak-valley TOU price curve of typical province.

### 5.2. Data Set

We chose 867 real customers' daily power usage data in China between 1st March 2016 and 31st March 2016. Customers were divided into four categories according to daily average electricity consumption, including less than 10 kWh, between 10 kWh and 20 kWh, between 20 kWh and 50 kWh, and between 50 kWh and 100 kWh as shown in Figure 5.

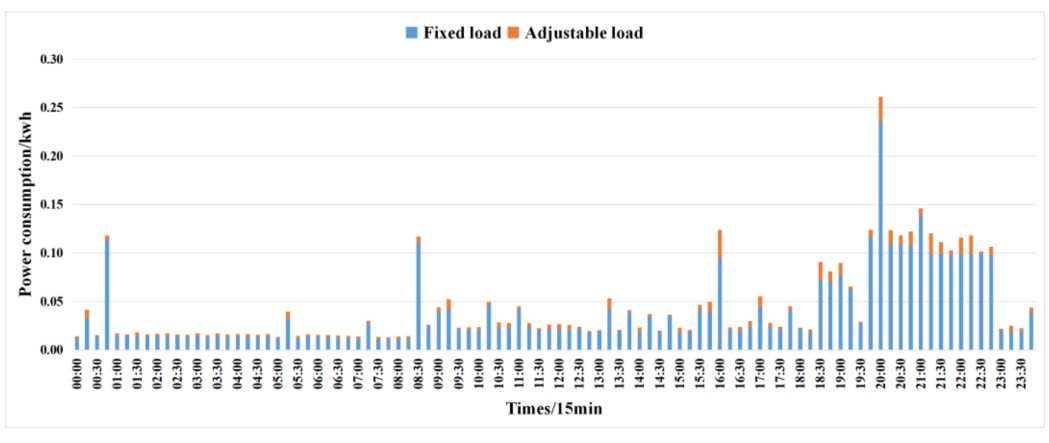

**(a) Category A**

**Figure 5.** *Cont.*

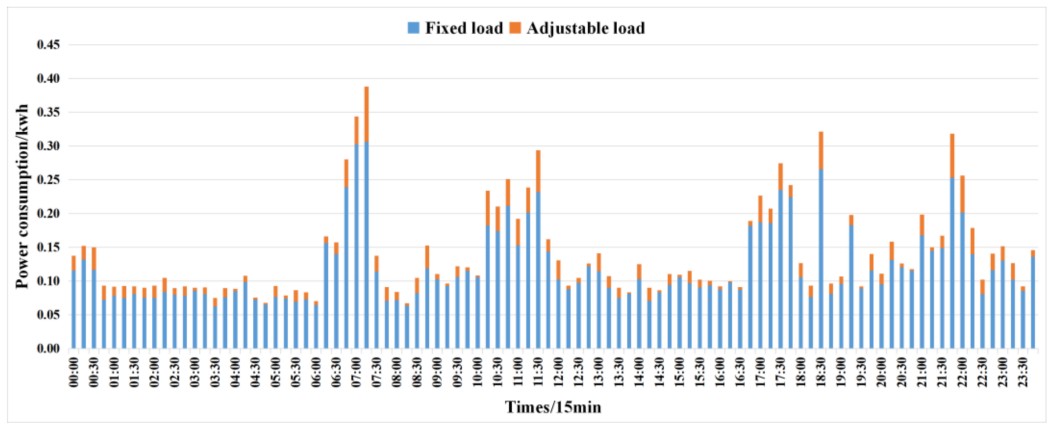

**(b) Category B**

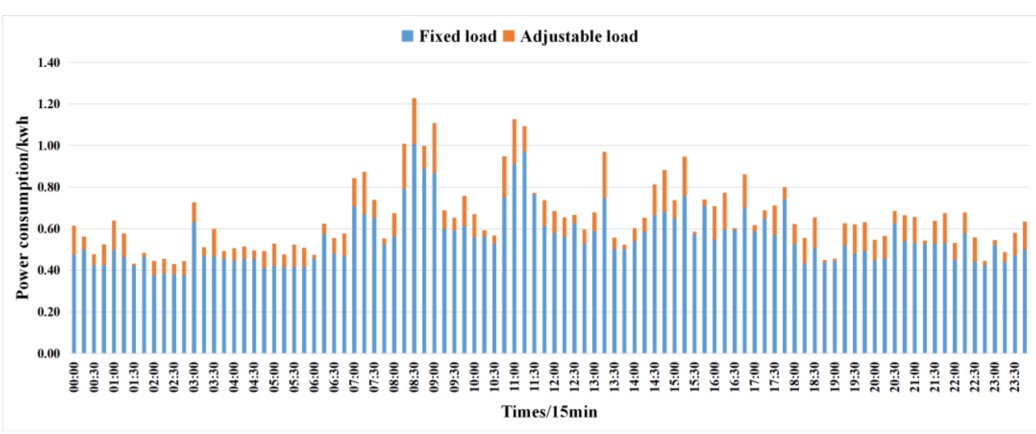

**(c) Category C**

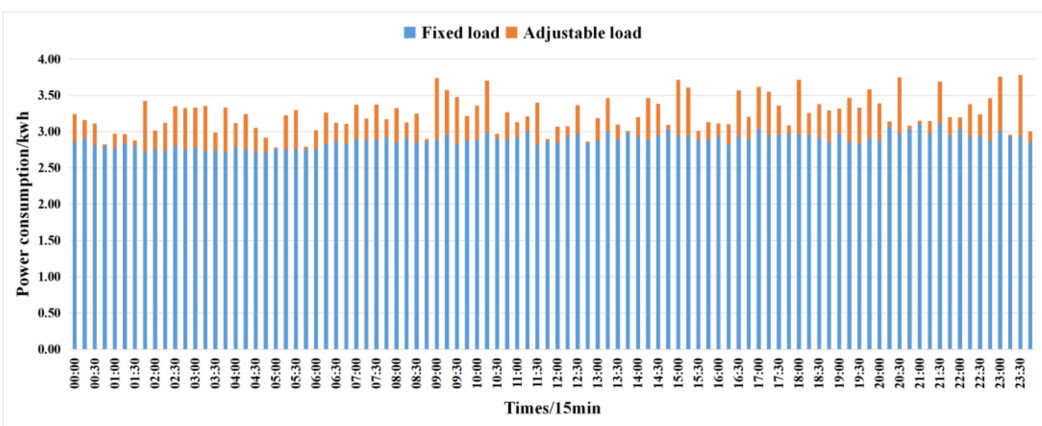

**(d) Category D**

**Figure 5.** Electricity consumption curve of typical customers.

The data set consists of the metering data from a widely installed smart meter. Considering the need for normal power consumption, the adjustable load to be used in demand response cannot be the whole load of customers. In the paper, we divided it into two parts: fixed load and adjustable load power consumption.

Fixed load is a basic electricity usage that does not decrease with the price adjustment. In fact, the power grid company and service provider use at least a month of the power consumption data in the past to ensure and evaluate the fixed load of customers. Unlike

fixed load, the adjustable load is affected by electricity price. Usually, we regard the air conditioner and heater as the mainly adjustable load. The work about how to ensure the fixed load and adjustable load has already been researched in many existing papers [47]. The detailed discussion and analysis of the divided method will not be presented in this paper.

### 5.3. Model Training

In the paper, Q-learning, DQN and DDPG models were training to generate the retail price. As the training processes were similar, the loss value is decreasing to keep stable and accumulative rewards keep increasing until they converged. It should repeat training processes many times to get a good result. A typical training process for DRL model is as shown in Figure 6, which is the model used to generate the retail price based on DDPG.

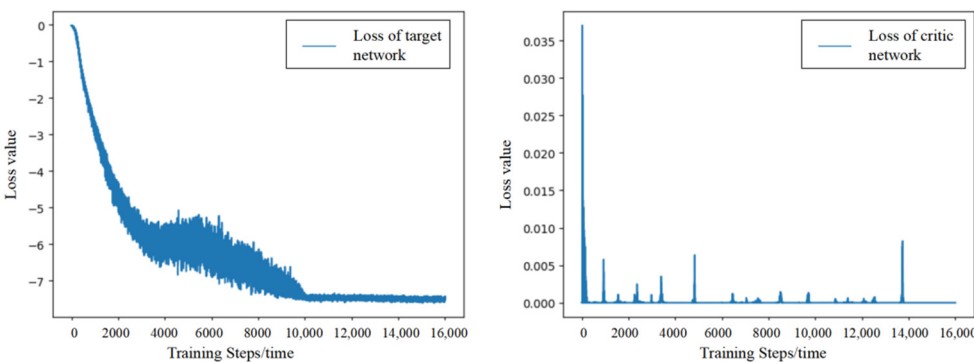

**Figure 6.** Training curve of optimal model.

### 5.4. Analysis of Retail Price Strategy

The comparison between different retail price curves by using Q-learning, DQN, DDPG models, and peak-valley TOU price are shown in Figure 7, and the service provider's profit and customer's cost for four customer groups with the unit "¥" which is China's currency yuan were shown in Table 3 based on different RL models.

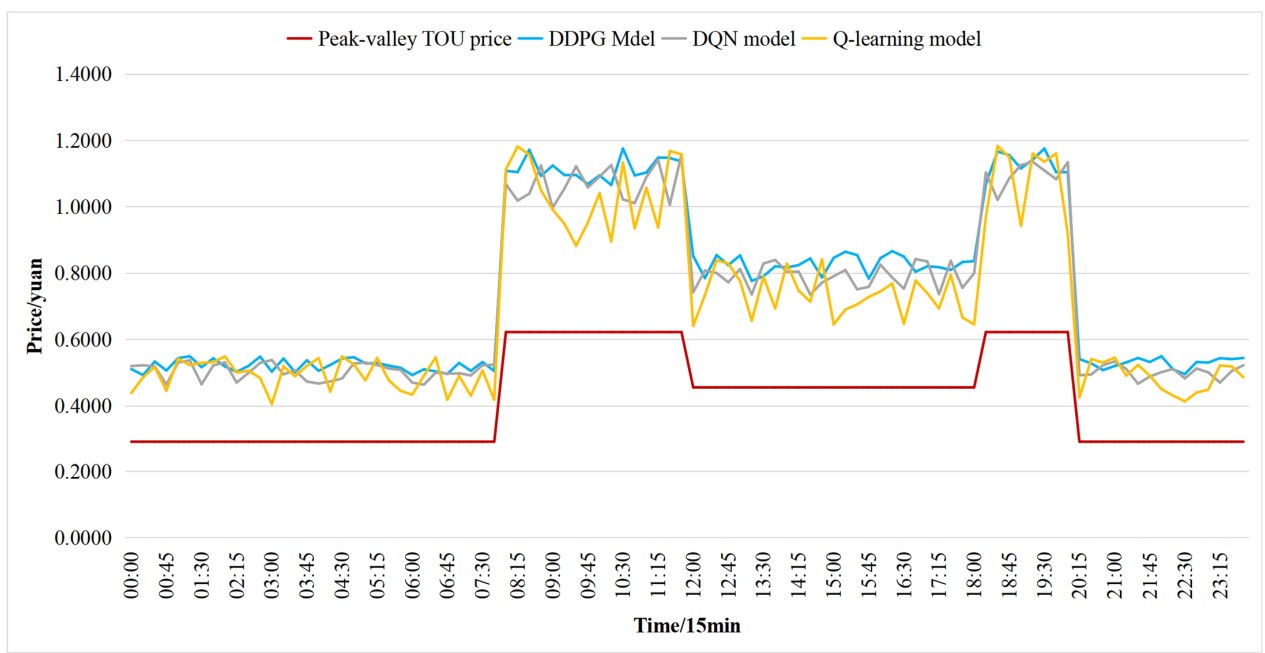

**Figure 7.** The retail price strategies based on different models.

**Table 3.** The Comparison of Different DRL Models.

| | | Q-Learning | DQN | DDPG |
|---|---|---|---|---|
| Group 1 | Service provider's profit | ￥ 96.12 | ￥ 103.24 | ￥ 242.53 |
| | Customer's cost | ￥ −1.11 | ￥ −3.26 | ￥ 10.31 |
| Group 2 | Service provider's profit | ￥ 80.22 | ￥ 125.0 | ￥ 260.41 |
| | Customer's cost | ￥ −2.32 | ￥ −1.21 | ￥ 20.76 |
| Group 3 | Service provider's profit | ￥ 108.31 | ￥ 150.51 | ￥ 272.08 |
| | Customer's cost | ￥ −1.54 | ￥ −2.5 | ￥ 22.43 |
| Group 4 | Service provider's profit | ￥ 96.31 | ￥ 135.51 | ￥ 226.08 |
| | Customer's cost | ￥ −1.50 | ￥ −2.42 | ￥ 13.41 |

Because of the range limit, it can be seen that the retail price converges to the value about twice the wholesale price, and the DDPG model has a higher and more stable performance. The retail prices generated by the DDPG model fluctuate within 5% while the retail prices generated by other algorithms fluctuate about 10% or 15%.

As the execution effect of the DDPG model, the actual power consumption is shown in Figure 8 compared with the adjustable load for different groups of customers. The DDPG model could give a suitable retail price for guiding customers to actively participate in the 85–95% percent of demand response programs.

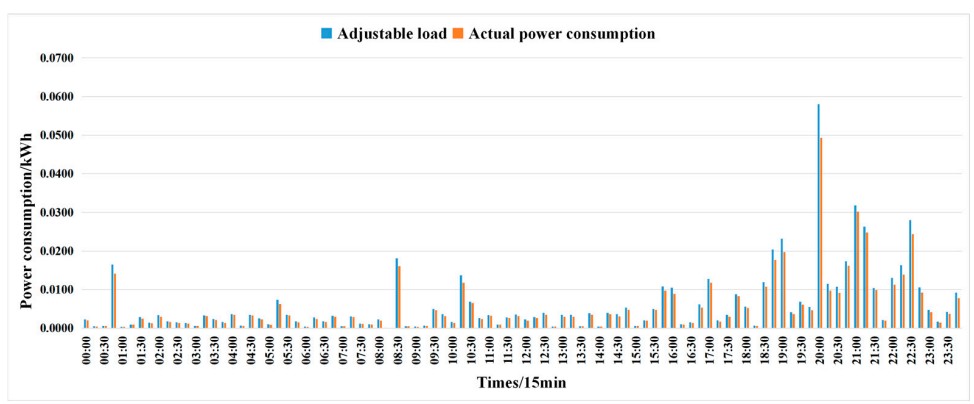

**(a) Category A**

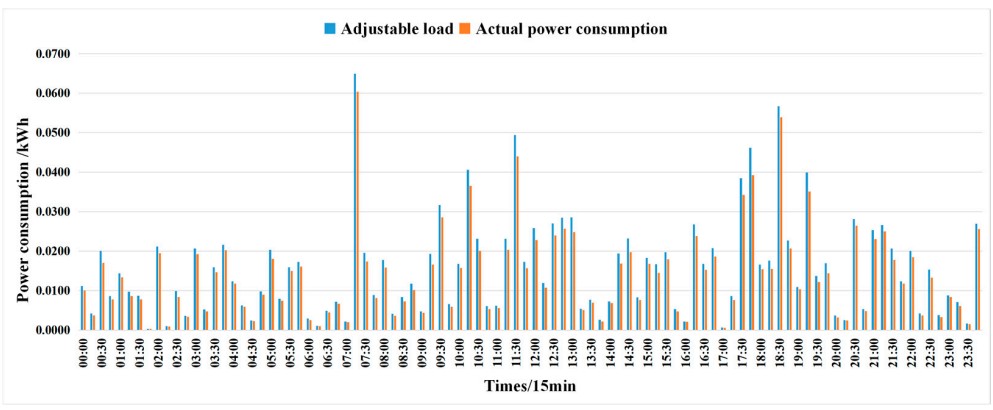

**(b) Category B**

**Figure 8.** *Cont.*

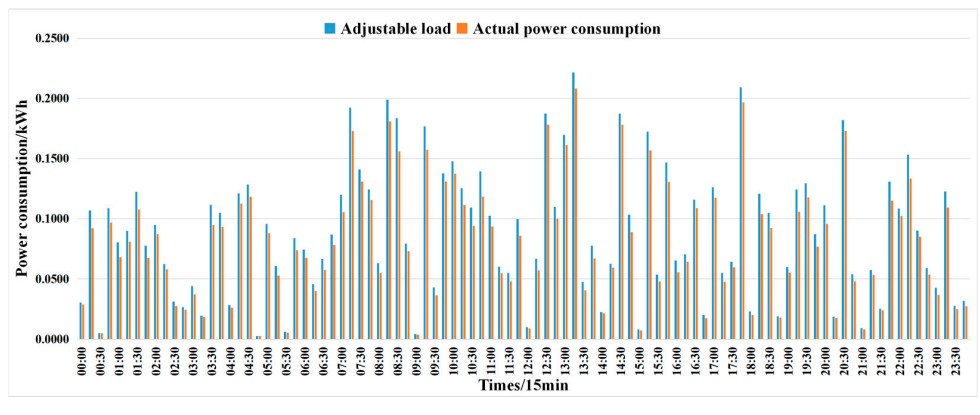

**(c) Category C**

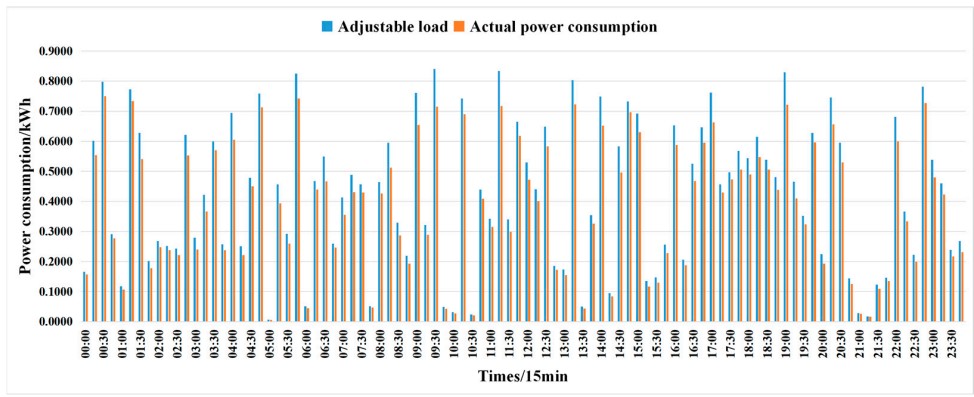

**(d) Category D**

**Figure 8.** The execution effect of DDPG model on four types of customers.

## 6. Conclusions

With the development of new electric power system reform in China, traditional centralized and unified planning management mode of power grid companies has struggled to meet the needs of development. How to strengthen the interaction between the customers and realize the further optimization of resource allocation have become key research directions. However, the interaction pattern is too complex to build a mathematical or physical model which could completely describe the process. DRL have attracted a great deal of attentions in recent years because of its advantages as an optimal model that can study the interactive experience in a model-free environment. A deep deterministic policy gradient model is proposed to automatically set retail prices for service providers in this paper, which could also realize the maximization of both power sales and power consumption. Obviously, the DDPG model is better for the deterministic policy when compared with Q-learning and DQN algorithms and has improved the training stability and convergence efficiency.

For future studies, further optimization of teamwork in the retail price strategy should guide the customers to actively participate. There are several future works needed:

(1) In this paper, the optimal model of retail price generation strategy depends on the electricity consumption every 15 min; a shorter time scale should be introduced in the future, such as one minute or in seconds.

(2) Determine if DRL model is suitable to deal with uncertain interaction process, and how to design the reward function has always been an important issue in modeling.

(3) Present research focuses on a simulation system that consists of simplified parts. It could be applied in a real situation in the future in the process of electric power system reform in China.

**Author Contributions:** Conceptualization, K.W.; Data curation, C.D.; Funding acquisition, C.D.; Methodology, C.D.; Software, C.D.; Writing—original draft, C.D.; Writing—review & editing, C.D. All authors have read and agreed to the published version of the manuscript.

**Funding:** This research received no external funding.

**Institutional Review Board Statement:** Not applicable.

**Informed Consent Statement:** Not applicable.

**Conflicts of Interest:** The authors declare no conflict of interest.

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
