# Peer review of "Residential Demand Response Strategy Based on Deep Deterministic Policy Gradient"

_processes, doi:10.3390/pr9040660_

Round 1

Reviewer 1 Report

Literature review can be improved giving more details for each of the references, for example some of the results of  other studies. In section 2 there are many related studies that are presented, however there is not a clear comparison with this study. I propose the formulation of paragraph which is going to give the differences of your study in order to stress the  novelty . Figure 7 and table 3 do not have the units of the presented magnitudes. In Line 370 is referred the that were training three models, however in section 4 there are presented only the proposed DDPG model. I think that conclusions must be improved with comments of the results for the comparisons of different models and for the optimal model. Finally, in conclusions it is important to add a sentence for the impact of this study which stresses its novelty.

Reviewer 2 Report

The authors did a great job.

Reviewer 3 Report

The article's idea is generally good, but there are many issues to be addressed.

Introduction: please rephrase and make more clear the statements on lines 35 - 37. Also provide citations for the shown data.

You are using roman figures when referring the sections (pages 100-104), but the respective sections are using arabic figures in the title of the sections.

It is not clear in Fig. 1 what "Grid" means. Is it the Power grid company? Also, please check the directions of the arrows in Fig. 1, currently some of them don't make sense (e.g. in your picture demand goes from the grid to the service provider).

Section 5.2 - Data Set: Please detail the criteria on which the five customers categories were chosen.

Please use captions and provide details for each of the two charts in Fig. 6.

Section 5.4 - Detail on the choice of the four Groups that are mentioned in Table 3. Why four groups and what was the criteria for them?

Please add more comments in the Conclusions section, based on the data that you provided. Especially, please comment thoroughly on the results presented in Fig. 7: which model is better, why and how can it be implemented in real-world environment?

Please provide full details on the quantities on both axes in all the charts and the respective units. Please mention the units correctly.

I suggest English proofreading to be considered.

Round 2

Reviewer 3 Report

Thank you for the updates. I think you addressed the initial comments in an adequate manner. 

However, I strongly suggest a proper English proofreading before the final submission.
